# Protective Factors for Vocal Health in Teachers: The Role of Singing, Voice Training, and Self-Efficacy

**DOI:** 10.3390/ijerph22071018

**Published:** 2025-06-27

**Authors:** Nora Jander, Nico Hutter, Thomas Mueller, Anna Immerz, Fiona Stritt, Louisa Traser, Claudia Spahn, Bernhard Richter

**Affiliations:** Freiburg Institute for Musicians’ Medicine, University of Music Freiburg, Medical Center and Faculty of Medicine–University of Freiburg, Freiburg Centre for Music Research and Teaching, Elsässerstr. 4 d, 79110 Freiburg, Germany; nora.jander@uniklinik-freiburg.de (N.J.); nico.hutter@uniklinik-freiburg.de (N.H.); anna.immerz@uniklinik-freiburg.de (A.I.); fiona.stritt@uniklinik-freiburg.de (F.S.); louisa.traser@uniklinik-freiburg.de (L.T.); claudia.spahn@uniklinik-freiburg.de (C.S.); bernhard.richter@uniklinik-freiburg.de (B.R.)

**Keywords:** vocal health, teacher well-being, singing, voice training, self-efficacy, prevention, protective factors

## Abstract

Voice disorders occur frequently in schoolteachers. The aim of the present cross-sectional study involving 124 German teachers was to investigate whether singing, voice training, and high self-efficacy are protective factors for vocal health. Furthermore, vocal self-concept was examined as a potential mediator explaining this relationship. Participants were assigned to the cases group if they had a clinically significant finding in voice examinations consisting of video laryngoscopy (VLS), auditory assessment (RBH), and the Voice Handicap Index (VHI) were assigned to the cases group. Psychosocial assessments comprised questions about singing activities and participation in voice training as well as validated questionnaires regarding self-efficacy (LSWS) and vocal self-concept (FESS). Group comparisons and mediation analyses were conducted. Analyses revealed a decreased risk of voice problems for teachers who sing regularly (OR: 0.442, *p* = 0.038). Furthermore, the absence of voice problems was associated significantly with higher self-efficacy ratings (t(113) = 1.71, *p* = 0.045). Both associations were mediated by vocal self-concept ratings (singing: ab = −0.422, 95%-CI [−1.102, −0.037]; self-efficacy: ab = −0.075, 95%-CI [−0.155, −0.022]). Participation in voice training in the past did not reduce the risk of voice problems significantly. The presented data suggest that regular singing and self-efficacy should be promoted in health care and prevention programs. Since no impact of sporadic participation in voice training activities on the occurrence of voice problems was found, efforts regarding the transfer of regular vocal exercises into daily life need to be intensified.

## 1. Introduction

Teachers belong to the group of speaking professions [1,2], as their professional activities rely heavily on vocal communication across various aspects of their daily work. Due to the intensive daily use of their voice, teachers face a higher risk of developing voice disorders compared to individuals in other professions [3,4]. According to a review, the risk of voice disorders among teachers is two to three times higher compared to the general population [5]. Nusseck and colleagues report that more than 50% of teachers suffer from voice problems at least once in the course of their professional career [6]. These findings are in line with another study, in which 64.9% of teachers exhibited a high level of vocal strain [7]. In addition, voice problems of teachers are associated with reduced cognitive and memory performances in students [8,9,10,11]. Furthermore, the existing literature highlights a frequent co-occurrence of vocal disorders and psychosocial problems [12,13,14,15,16,17] that lead to increased teacher absences and subsequently increased economic costs for the health care system [18]. Therefore, from a health care perspective, it is mandatory to identify factors reducing the risk of developing voice problems and to provide interventions as well as prevention programs to strengthen the health of teachers in the long term.

Singing in a choir positively affects the quality of life and physical well-being in healthy adults [19]. Singing has also been shown to have positive effects on various health outcomes in older adults and those with mental health problems, lung disease, stroke, and dementia [20]. A positive preventive impact of singing on the vocal function of the elderly is reported [21]. Furthermore, a systematic review suggests that singing as an intervention for older adults can improve respiration and respiratory/phonatory control [22]. In a study of children, significantly fewer voice disorders were found in a singing group than in a non-singing group [23]. To the best of our knowledge, there are no studies examining the effects of singing on vocal health in teachers. Nevertheless, we hypothesize that regular singing (i.e., choir, band, and lessons) might exert a positive impact on the vocal health of teachers as well.

Voice training interventions positively affect vocal health with regard to self-reported vocal symptoms in adult patients and teachers with functional dysphonia, as summarized in a systematic review and meta-analysis investigating direct and indirect voice therapy as well as other treatments such as pharmacological treatment and vocal hygiene instructions [24]. While another meta-analysis of six randomized clinical trials focusing on preventive impact found no effect of direct, indirect, or combined voice training on the prevention of voice disorders in healthy adults [25], recent evidence from a study of teachers who participated in voice training sessions covering economical vocal techniques, individual coaching, and information on improving room acoustics and noise management suggests that voice training reduces voice problems and improves vocal resilience [7]. Moreover, positive long-term effects of voice training with regard to an increased awareness of voice use and mental well-being in teachers are reported [17].

Self-efficacy as a part of the individual’s self-concept refers to a person’s confidence to carry out desired actions based on their own skills and to be able to cope successfully with challenging situations [26]. Teachers often encounter challenging situations in their professional lives that can lead to voice problems [4,27]. In addition to behavioral issues in the classroom [14,28], factors such as large class sizes [14,28,29], speaking in an amplified tone [14], noise [14,28,29], and long lesson times [29] may contribute to voice problems. In the school context, self-efficacy has been shown to be a powerful resource regarding the mental health of teachers [30,31,32]. Studies point out that low self-efficacy among teachers is associated with high levels of stress and reduced enjoyment of professional activities [30,31]. Trainee teachers with higher self-efficacy exhibit greater psychological well-being, more optimism, and less stress [31,32]. Moreover, higher self-efficacy of teachers seems to be associated with better teaching quality [31]. Although there is no data on the direct relationship between self-reported self-efficacy and vocal health yet, we assume that higher self-efficacy beliefs may reduce the occurrence of voice problems in teachers. This relationship may be attributed to the fact that self-efficacy enhances psychological well-being, which, in turn, is associated with improved vocal health [13,14,15,16,17]. Additionally, as higher self-efficacy generally implies better coping with challenging situations, teachers with higher self-efficacy might possess the ability to maintain vocal health even in the case of beginning voice problems.

As evidenced by the aforementioned studies, regular singing, voice training, and high self-efficacy are likely to contribute to long-term vocal health. However, little is known about possible mechanisms of action so far. One construct that may play a particularly significant role in this context is vocal self-concept, which refers to an individual’s relationship with their own voice and their awareness of its use.

Raising awareness regarding the vocal strains accompanying the teaching profession and a conscious, positive relationship with one’s voice might be requirements in maintaining vocal well-being for teachers in the long term. This association is suspected because face-to-face and blended-learning voice training interventions that include efforts to increase teachers’ self-efficacy have shown significant improvements not only in vocal health, but also in how comfortable teachers feel with their voice, how they can express themselves emotionally with their voice, and how carefully they handle their voice [17,33,34]. Therefore, the positive effects of voice and self-efficacy training on vocal health might be substantially affected by improvements in the vocal self-concept. For singing, similar studies do not exist yet. However, we propose a comparable mechanism of action, suggesting that the positive effect on vocal health may stem from enhanced vocal self-monitoring skills in individuals who sing regularly. These individuals are likely to have a more refined auditory perception of deviations in vocal parameters and speech characteristics—such as loudness, modulation, pitch, phonation, and articulation—which enables them to recognize and address these deviations more effectively. This may contribute to a better vocal self-concept. Consequently, they may be better equipped to prevent or counteract harmful patterns of voice use.

Thus, the present study aimed to investigate whether singing, voice training, and self-efficacy are indeed protective factors for vocal health in teachers. Furthermore, we examined the impact of vocal self-concept on the association between protective factors and voice problems.

## 2. Materials and Methods

This cross-sectional study is based on a convenience sample of 215 German schoolteachers from Baden-Württemberg who took part in a blended-learning prevention program for vocal and mental health (December 2021 to June 2024). During this period, three rounds of blended learning training were carried out, comprising an online kick-off event, eight e-learning units to be completed independently, and two face-to-face events. In the first round of the program (December 2021 to July 2022), 83 teachers participated. In the second round (October 2023 to January 2024), 71 teachers participated, and in the third cohort (March to June 2024), there were 61 participating teachers. The program is open to teachers of all school types. At the very beginning of each round, data were collected via questionnaires, and all participants were offered voice examinations. The sample of this study consists of the group of participants who took part in the voice examination (*n* = 151) and completed the questionnaires (*n* = 124) before taking part in the blended learning training. Hence, further data analyses refer to this subset of participants (82%, see Figure 1). Participants gave their informed consent to collect data via questionnaires and the voice examination in accordance with the ethical committee of the University Clinic Freiburg (11 February 2021: 21-1062).

The 124 participants with complete data sets were divided into cases (*n* = 84) and controls (*n* = 40). Participants were assigned to the cases group if they had at least one clinically significant finding in any of the following three diagnostic measures: (a) Voice Handicap Index (VHI; severity level 1 or higher), (b) video laryngoscopy (VLS), or (c) roughness, breathiness, and hoarseness scale (RBH) (R1B0H1 or R0B1H1 or higher). All other participants were assigned to the control group.

### 2.1. Voice Examination

The voice examinations consisted of a video laryngoscopy (VLS) by a physician, an auditory assessment of the voice by a physician and a speech language pathologist with regard to roughness, breathlessness, and hoarseness (RBH), and the Voice Handicap Index (VHI). Teachers with clinically significant findings were recommended to undergo further voice diagnostics in the outpatient clinic of the Freiburg Institute for Musicians’ Medicine. Voice examinations were conducted according to the guidelines of the European Laryngological Society [35] and the current German guideline of the Association of the Scientific Medical Societies in Germany [36].

#### 2.1.1. Laryngoscopy

The laryngoscopy was performed transnasally or transorally by a physician and recorded with the program DiVAS by Xion (2.8, Berlin, Germany). Two physicians assessed the VLS results for organic abnormalities and diseases such as mass lesions (e.g., polyps, cysts, edema, and scars), vocal fold paralysis, infections, neoplasms, or varices.

#### 2.1.2. Roughness, Breathiness, and Hoarseness (RBH)

The RBH test [37] is used to evaluate voice quality. Both a physician and a speech therapist assessed the parameters of roughness, breathiness, and hoarseness. The value range of the RBH system extends from no disturbance (0) to minor disturbance (1), moderate disturbance (2), and severe disturbance (3), in terms of roughness (R), breathiness (B), and hoarseness (H). A clinically noticeable RBH was determined if at least one of the two examiners rated an RBH value of more than 0.

#### 2.1.3. Voice Handicap Index (VHI)

The German version of the Voice Handicap Index was used to determine the subjective limitation of quality of life caused by the voice [38]. This standardized questionnaire for determining a voice disorder contains 30 items. In the present project, a short version consisting of 12 items with a five-level response scale ranging from “never” (zero) to “always” (four) was used. The measure of internal consistency of the VHI-12, Cronbach’s α, is 0.91. The calculation of the VHI-score was performed by simple summation. The score was then converted into one of four severity levels: severity level 0: no voice disease (score of 0–7), severity level 1: low restriction (8–14), severity level 2: moderate restriction (15–22), severity level 3: high restriction (23–48). A severity level higher than 0 was defined as an abnormal value [39].

### 2.2. Psychosocial Questionnaires

Next to information on age, gender, and work-related aspects, the teachers were asked whether they sing regularly (e.g., choir or singing lessons) or whether they currently participate or had ever participated in voice training activities. The questionnaires described below were used in the study.

#### 2.2.1. Self-Efficacy in Teachers (Lehrer-Selbstwirksamkeits-Skala (LSWS))

The concept of self-efficacy asks for a person’s assessment of their ability to cope with difficulties and obstacles in everyday life. This concept was transferred to the teaching profession and measures the job-specific belief in competence to meet the tasks of a teacher [40]. The standardized and validated scale used here contains 10 items with a four-level response scale ranging from “strongly disagree” (one) to “strongly agree” (four). The measure of internal consistency of the LSWS, Cronbach’s α, is 0.80 [41]. The calculation of the LSWS-score was performed by simple summation after inverting item 9. Examples of items are: “I am confident in my ability to successfully teach even the most challenging students” or “I can implement innovative changes even in the face of skeptical colleagues”.

#### 2.2.2. Vocal Self-Concept (Fragebogen zur Erfassung des Stimmlichen Selbstkonzepts (FESS))

This German-language questionnaire allows a self-assessment of one’s voice in relation to the self-concept. The questionnaire consists of 17 items with a three-dimensional factor structure and a five-level response scale ranging from “does not apply” (one) to “applies very much” (five) [42]. For our study, we used the dimensions “relationship with the own voice” with 6 items (Cronbach’s α = 0.83; example: ”I know my voice very well”) and “awareness of voice use” with 6 items (Cronbach’s α = 0.79; example: “I use my voice consciously”) because they reflect aspects relevant to vocal health such as the perception of the own voice and the conscious use of the voice. The score for each dimension was calculated by determining the mean value of the six items. Since the dimension “voice and emotion” shows limited reliability (Cronbach’s α = 0.66) and does not appear to be linked to vocal health in terms of its content, this scale is omitted from the analyses.

### 2.3. Statistical Methods

The data analysis was carried out using SPSS (version 29) and R (version 4.4.1). Differences between the two groups were examined using t-tests for independent samples for interval-scaled variables (one-sided significance tests; significance level α < 0.05). Group comparisons for nominal-scaled variables were conducted using Pearson chi-square tests (significance level α < 0.05), and odds ratios (ORs) with corresponding confidence intervals were calculated. The prerequisites were checked in each case (check for homogeneity of variance (Levene’s test) and for low expected frequencies). To assess possible explanations of the protective effect of singing, voice training, and/or self-efficacy on the voice, we conducted mediation analyses using the PROCESS macro by Hayes [43], which uses logistic regression, yielding unstandardized path coefficients for indirect effects. Bootstrapping with 10,000 samples together with heteroscedasticity-consistent standard errors [44] were employed to compute the confidence intervals and inferential statistics. Effects were deemed significant when the confidence interval did not include zero. Mediation analyses were conducted if a significant association between investigated protective factors and vocal health was found.

## 3. Results

Regarding socio-demographic variables, there are no significant differences between cases (*n* = 84) and controls (*n* = 40; Table 1) for the 124 teachers who participated in voice examination and questionnaire survey. Of the 124 participants, 71 teachers reported a voice problem by VHI, 37 of the teachers had a clinically significant RBH and 16 a significant finding in the laryngoscopy.

### 3.1. Analyses Regarding Protective Factors

A significant difference was observed, with teachers who engage in regular singing exhibiting a notably lower risk of experiencing voice problems (χ2 = 4.29, df = 1, *p* = 0.038, OR: 0.442, 95%-CI [0.203, 0.965]). Pearson–Chi-Square tests revealed that 48% of the participants who do not suffer from voice problems sing regularly whereas just 29% of the participants who suffer from voice problems sing regularly.

Furthermore, 55% of participants without voice problems reported engaging in voice training, while 52% of those with voice problems indicated participation in voice training. This difference was not statistically significant (χ2 = 0.08, df = 1, *p* = 0.785, OR: 0.9, 95%-CI [0.423, 1.917]).

Regarding self-efficacy in teachers, a significant difference was found between the groups: teachers without voice problems rated their self-efficacy significantly higher than those with voice problems (t(113) = 1.71, *p* = 0.045).

### 3.2. Additional Analyses

#### 3.2.1. Vocal Self-Concept (FESS)

The two dimensions “relationship with the own voice” (t(113) = 4.05, *p* < 0.001) and “Awareness of voice use” (t(113) = 1.71, *p* = 0.045) of the FESS questionnaire show significant differences between the two groups with control group ratings higher than those in the cases group.

#### 3.2.2. Singing and Vocal Self-Concept

The evaluations of the FESS questionnaire show that teachers who sing regularly have significantly higher values in the dimensions “relationship with the own voice” (t(113) = −2.87, two-sided *p* = 0.005) and “awareness of voice use” (t(113) = −2.35, two-sided *p* = 0.022) compared to teachers who do not sing (Table 2).

Additionally, a mediation analysis was performed to analyze whether the direct path (regular singing predicting the absence of voice problems) would be mediated by the relationship with the own voice and the awareness of using one’s voice. We found that the relationship between regular singing and the absence of voice problems in teachers is fully mediated by the relationship with the own voice and the awareness of using one’s voice, indirect effect ab = −0.422, 95%-CI [−1.102, −0.037].

#### 3.2.3. Self-Efficacy in Teachers (LSWS) and Vocal Self-Concept

In terms of self-efficacy in teachers there is a similar pattern: there is a significant correlation (Table 3) between self-efficacy and the FESS-dimensions “relationship with the own voice” (two-sided *p* < 0.001) and “awareness of voice use” (two-sided *p* = 0.002).

Similarly to the mediation analysis regarding singing, we also performed a mediation analysis to analyze whether the direct connection between self-efficacy and voice problems would be mediated by the relationship with their own voice and awareness of voice use. We found that the relationship between self-efficacy and voice problems in teachers is fully mediated by the relationship with the own voice and the awareness of voice use (indirect effect ab = −0.075, 95%-CI [−0.155, −0.022]).

## 4. Discussion

Teachers are at high risk of developing voice disorders due to the vocal demands of their profession [3,4]. The present study provides data on German schoolteachers exploring the role of singing, voice training, and self-efficacy as protective factors for vocal health. The findings show that singing and high self-efficacy reduce the risk of voice problems in teachers, whereas sporadic voice training activities had no effect on vocal health. These results have to be considered in the development and promotion of targeted interventions regarding the prevention of voice problems in teachers.

The finding of a significant portion of teachers with voice problems in this study (67.7%) is consistent with other studies that report a high percentage of teachers with voice problems [7]. The high number of teachers with voice problems underlines the vocal demands of teachers’ daily lives and highlights the need for targeted health services in order to prevent voice disorders and maintain vocal health in the long term. As the teachers participating in this study subsequently took part in a prevention program on vocal and mental health voluntarily, it can be assumed that they are not only interested in this topic, but are already experiencing significant stress in this area. Therefore, it can be assumed that this is not a pure prevention group, but also at least partly an intervention sample, and the results must be interpreted against this background.

The present study is the first to show evidence that regular singing, such as singing in a choir or taking singing lessons, functions as a protective factor for vocal health among teachers. Even the increased mechanical strain of regular singing does not counteract the obvious positive effects on vocal health. This might be due to a direct positive effect of singing through physiological changes in the voice, which contribute to maintaining and fostering vocal health. Yet, the analyzed data suggests another possible and indirect mechanism of action: regular singing might lead to better vocal health through higher values in vocal self-concept, i.e., positive relationship with the own voice and high awareness regarding voice use. This finding is in line with a study by Nusseck et al., which shows that a positive relationship with one’s voice is associated with a decrease in self-reported voice problems [17].

Thus, the perception of one’s voice seems to play a crucial role for vocal health, since raising awareness regarding one’s voice and recognizing vocal demands and burden are indispensable prerequisites to focus on vocal hygiene and to train the voice intentionally in order to maintain vocal health in the long term. Regular singing might contribute to this by improving the auditory perception of one’s voice. Vocal deviations in voice parameters and speech characteristics (i.e., loudness, modulation, pitch, phonation, and articulation) can be more easily perceived and identified. Subsequently, these increased self-monitoring skills allow for an adaptation of voice parameters and speech characteristics that could be harmful. Furthermore, regular singing may improve vocal neuromotor control, facilitating the adaptation of the voice to the various situations teachers face in their daily work. The individual components of the mechanism of action should be investigated in more detail in subsequent studies.

Sporadic voice training did not exhibit a similar effect on vocal health, and our assumption was not confirmed in this sample. Most participants reported sporadic voice training activities that mostly occurred years ago (e.g., as part of their university studies), and no participant reported regular voice training activities. We assume that the positive effects of participation in singular voice training diminish over time. Studies examining the protective effect of one-time voice training (e.g., [25]) mainly focused on immediate protective effects on the voice. Yet, regularly engaging with one’s voice seems to be an important part of the protective effect, as singing in a choir or taking singing lessons brings with it a certain regularity, which sporadic voice training in the past cannot convey. Therefore, future studies should focus on the protective effects of regular voice training and on the impact of frequency and intensity of voice training. Additionally, the differential effects of specific exercises of voice training programs should be examined more closely to determine which aspects of voice trainings exert a substantial effect on vocal health.

The study underlines the influence of self-efficacy in protecting the health of teacher’s voices. Teachers who reported higher self-efficacy scores were less likely to experience voice issues, supporting the notion that psychological factors play an important role in vocal health [13,14,15,16,17]. Since teachers use their voice as one of the most important everyday tools in their work life, the skill of managing challenging situations effectively in general (i.e., high self-efficacy) might also be helpful in coping with voice difficulties. In a study of 291 teachers a considerable positive relation between proactive coping and self-efficacy was reported [45]. Based on these findings, one can assume that self-efficacy might be a resource itself for the health of teachers’ voices. Moreover, regular singing could have a positive influence on teachers’ confidence in their own voice by improving their self-monitoring skills and their ability to adapt their voice consciously to various situations in their daily work. The exact nature of this connection could be subject to further research.

Furthermore, the results of the study reveal a close association between self-efficacy and vocal self-concept in teachers, suggesting another explanation for the protective effect of self-efficacy on voice health: higher self-efficacy is associated with an increased awareness regarding voice issues and a more effective and healthy utilization of the voice, which in turn leads to better vocal health. The vocal self-concept aspect referring to the awareness of voice use can also be regarded as self-efficacy, since proactive utilization of one’s voice and coping with voice difficulties are inherent self-efficacy skills. Thus, a high level of general self-efficacy is associated with a positive vocal self-concept and subsequently leads to a higher sensitivity to the health of one’s voice. Of course, we cannot derive definitive implications about causality from our data. Further research is needed to establish a causal understanding of this association.

## 5. Conclusions

This study is the first to show that singing regularly and higher self-efficacy reduce the risk of voice problems in teachers. Thus, regular singing and strengthening self-efficacy should be considered and promoted in health care and prevention programs to foster long-term vocal health and to offer teachers self-strengthening opportunities to create a basis for a healthy professional practice. Both protective factors might exert their positive impact through a positive vocal self-concept in terms of raising awareness regarding a more conscious use of one’s voice. Furthermore, regular singing not only increases teachers’ vocal health but also increases quality of life and well-being [19] and might reduce absenteeism and subsequent health care costs. From a health care perspective, these positive effects should be put into practice, since singing is a low-threshold and easily accessible activity.

Moreover, the findings underscore the potential benefit of incorporating psychological components, such as self-efficacy training, into voice prevention programs for teachers. Specific training programs targeting the improvement of self-efficacy might exert a positive impact on sustaining and fostering vocal health of teachers and should thus be focused on in future research and health care programs.

Since no impact of sporadic participation in voice training in the past was found on the occurrence of voice problems, prevention programs have to focus on the transfer of regular voice training activities into teacher’s daily life. Further research is needed to confirm these findings and to explore additional protective factors that may contribute to the vocal well-being of teachers.

As the participants of this study stem from a convenience sample of teachers who participated in a prevention program for vocal and mental health, the generalizability of the results is limited. The cross-sectional design limits the ability to draw causal conclusions regarding the protective effects of singing and self-efficacy. Longitudinal studies with a larger and randomized sample are necessary to confirm these findings and to determine whether regular singing and a good self-efficacy lead to sustained improvements in vocal health over time. Although we believe the validity of the two used scales of the FESS (“relationship with the own voice” and “awareness of voice use”) to be sufficient, an improvement in the quality of the measurement instrument could further increase the interpretability and generalizability of the results of future studies.

## Figures and Tables

**Figure 1 ijerph-22-01018-f001:**
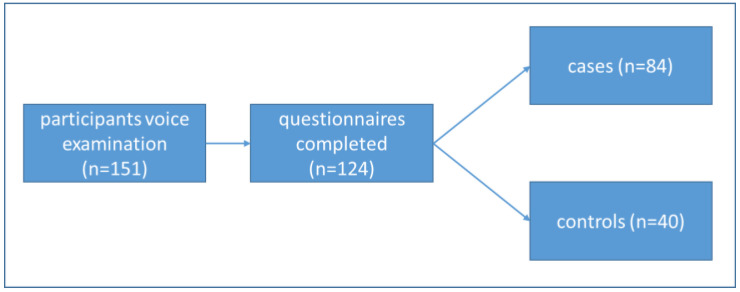
The flow chart illustrates the differentiation into two subgroups: cases and controls.

**Table 1 ijerph-22-01018-t001:** Socio-demographic data and psychosocial variables (*: *p* < 0.05; VHI: Voice Handicap Index; RBH: roughness, breathiness, and hoarseness Scale; FESS: questionnaire to assess vocal self-concept (Fragebogen zur Erfassung des stimmlichen Selbstkonzepts).

Completed Questionnaires (*n* = 124)	Case	Control
*n* (%)	84 (67.7%)	40 (32.2%)
**Voice examination**		
VHI severity scale > 0: *n* (%)	71 (84.5%)	0
RBH clinically noticeable: *n* (%)	37 (44.0%)	0
Significant laryngoscopic findings: *n* (%)	16 (19.0%)	0
**Socio-demographic data**		
Age: mean (SD)	45.24 (9.53)	45.28 (8.84)
Gender (female): *n* (%)	69 (82.1%)	37 (92.5%)
Part-time work: *n* (%)	49 (59.0%)	27 (67.5%)
Voice problems experienced before: *n* (%)	61 (80.2%)	26 (66.7%)
**Vocal self-concept**		
FESS: “relationship with the own voice”: mean (SD) *	3.11 (0.63)	3.63 (0.69)
FESS: “awareness of voice use”: mean (SD) *	3.01 (0.59)	3.23 (0.7)
**Protective factors**		
Singing regularly (in choir, band or singing lessons): *n* (%) *	24 (28.6%)	19 (47.5%)
Ever participated in voice training activities (e.g., education of voice physiology, voice exercises): *n* (%)	44 (52.4%)	22 (55.0%)
Self-efficacy in teachers (LSWS): mean (SD) *	28.99 (4.07)	30.31 (3.61)

**Table 2 ijerph-22-01018-t002:** FESS and regular singing (*: *p* < 0.05).

FESS-Dimensions	Singing	No Singing
FESS: relationship with the own voice: mean (SD) *	3.55 (0.78)	3.14 (0.59)
FESS: awareness of voice use: mean (SD) *	3.28 (0.72)	2.98 (0.55)

**Table 3 ijerph-22-01018-t003:** FESS and self-efficacy in teachers (*: *p* < 0.05).

FESS-Dimensions	Self-Efficacy (LSWS)
FESS: relationship with the own voice: Pearson r *	0.377
FESS: awareness of voice use: Pearson r *	0.284

## Data Availability

The data that support the findings of this study are available from the corresponding author upon reasonable request.

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
