# Peer review of "Protective Factors for Vocal Health in Teachers: The Role of Singing, Voice Training, and Self-Efficacy"

_ijerph, 2025, doi:10.3390/ijerph22071018_

Round 1
Reviewer 1 Report
Comments and Suggestions for Authors
I thank the authors for the opportunity to review this manuscript. It is a pleasure to contribute to the evaluation of a study that addresses such a relevant topic as the vocal care of teachers, particularly considering the vocal demands and challenges faced by this professional group. Studies that investigate predictive factors are essential to better understand effective intervention strategies for this population.
Below, I offer my comments with the intention of supporting the improvement of the manuscript.
Abstract
P1
L15-16: Mention the procedures that were performed in voice examinations.
L19-23: Insert all significant results to facilitate understanding of the findings of this study. Cite p values.
Introduction
P2
L48-58: In this section, it is important to include the relationship between singing and vocal self-monitoring, especially how it supports the control and adjustment of pitch and loudness. These parameters are essential in the vocal use of teachers during their professional activities. Another important factor is auditory perception. It is essential for singing, as it helps with tuning adjustment, voice modulation and correction of vocal deviations, which allows for vocal self-monitoring. It is important that these concepts are also incorporated into the discussion section.
L59-66: What kind of strategies were used? How long was the intervention?
L67-81: What challenging situations do teachers face in the professional context? Stress was mentioned, but issues related to the environment, noise, long working hours, allergens, vocal use, and student behavior are also factors that can affect the voice and vocal health.
Materials and Methods
P4
L135-138: What value range was used for evaluation? Another suggestion is to classify participants according to the type of dysphonia (compartmental or organic).
L139-144: What range of values ​​is used? What is the cutoff value of the instrument?
145-161: How are LSWS and FESS protocols evaluated? What range of values ​​is used?
165-167: Was the p-value considered =0.05 or <0.05?
P4-P5
Results
177-181: In general, I missed a more detailed description of the teachers who participated in this study, in relation to gender, age, length of service, classification of vocal deviation and dysphonia, among other factors that can help to better understand the profile of this population.
P7
Discussion
245-258: This paragraph should highlight that singing involves more refined vocal adjustments, enhanced auditory perception, and improved self-monitoring skills. Could these elements be contributing to participants’ vocal self-concept and self-efficacy? Might this explain the positive outcomes observed in the predictive values?
265-268 and 273-281: Teachers often show low adherence to traditional therapeutic processes. However, do they engage more readily with singing-based activities? Could this be related to their vocal self-concept or perceived self-efficacy?
P7-P8
282-292: I believe the discussion would benefit from the inclusion of a paragraph linking vocal neuromotor control (auditory and somatosensory feedback mechanisms) to the development of self-regulatory vocal skills. These skills directly influence vocal confidence and the perception of vocal competence, which are key aspects related to vocal self-efficacy and vocal self-concept.
Reviewer 2 Report
Comments and Suggestions for Authors
The article deals with the singing voice in the context of the vocal self-concept. The topic is original and in principle worthy of publication.
The methodology and structure of the study are conclusive, however, the questionnaire used for the vocal self-concept (Fragebogen zur Erfassung des stimmlichen Selbstkonzepts 155 (FESS)) does not fulfill the quality criteria in the area of validity, especially in the context of construct validity in the area of factor analysis.
Abstract: Information on the methodology, the sample size and the measuring instruments used is missing.
Study design: well-chosen structure. However, it appears to be part of a larger study, which is not discussed in detail.
Results: The results themselves are well presented.
Discussion: The interpretation of the results must be made against the background of the limitations of the individual measuring instruments and cannot be generalized. If the questionnaire is not able to capture the self-concept validly, it is questionable how meaningful they are. This should at least be discussed.
Reviewer 3 Report
Comments and Suggestions for Authors
To authors,
The manuscript, Protective Factors for Vocal Health in Teachers: The Role of Singing, Voice Training, and Self-Efficacy, is an interesting scholarly contribution providing some evidence to support the interaction between singing and improved psychological and vocal health in teachers. As acknowledged by the authors, the cross-sectional design does not provide strong evidence of causality. However, this research does provide important implications, that aspects of psychological health (self-efficacy and self-concept) that must be included and evaluated in pre-service teaching education as well as other professional disciplines such as vocal or instrumental music teaching education.
I found very minor errors in the text and propose the following questions and suggestions which might be considered for revision.
Abstract
The third sentence states this is a case-control study. To avoid confusion, could you state ‘cross-sectional study’ as you have done in the methods description (p.3, line 103).
Materials and methods
Examinations and standardized questionnaires appear appropriate for the study design. For the two German-language psychosocial questionnaires, would it be possible to include a short description of how the scale or measurement would indicate relative results of self-efficacy and self-concept. This would help the reader when interpreting the results.
Results
Second sentence, could you please change ‘cases’ for ‘participants’ (p.5, line 179).
Also, at the end of the last paragraph there appears some irrelevant text which needs to be deleted from the proof (p.6 lines 229-231).
Discussion and conclusions
The discussion and conclusions of this research are very promising. However, there is no discussion whether or not the total number of cases compared to half that number in the control group – does this uneven number impact interpretation of the results? Authors stated that this study of a convenience sample was limiting and should also consider sample bias of participants who previously attended a vocal training education course might also be more likely to have interest in the current study and to care about their health in general.
I enjoyed reading your well-written manuscript describing a well-planned research study. And I look forward to seeing further research from this team in the future.
